# A Novel Type of Pseudo-Decoupling Method for Two Degree-of-Freedom Piezoelectrically Driven Compliant Mechanisms Based on Elliptical Parameter Compensation

**DOI:** 10.3390/mi14112043

**Published:** 2023-10-31

**Authors:** Rongqi Wang, Xiaoqin Zhou, Haonan Meng, Baizhi Liu

**Affiliations:** Key Laboratory of CNC Equipment Reliability, School of Mechanical and Aerospace Engineering, Jilin University, Ministry of Education, Changchun 130022, China; xqzhou@jlu.edu.cn (X.Z.); menghn2022@163.com (H.M.); liubz22@mails.jlu.edu.cn (B.L.)

**Keywords:** piezoelectrically driven compliant mechanisms (PDCMs), pseudo-decoupling method, trajectory tracking accuracy, elliptical parameter compensation (EPC)

## Abstract

At present, a large number of two-degree-of-freedom piezoelectrically driven compliant mechanisms (2-DOF PDCMs) have been widely adopted to construct various elliptical vibration machining (EVM) devices employed in precisely fabricating functional micro-structured surfaces on difficult-to-cut materials, which have broad applications in many significant fields like optical engineering and precision manufacturing. For a higher precision of conventional 2-DOF PDCMs on tracking elliptical trajectories, a novel type of pseudo-decoupling method is proposed based on phase difference compensation (PDC). With finite element analysis (FEA), the dependences of elliptical trajectory tracking precision on PDC angles will then be investigated for optimizing PDC angles under different elliptical parameters. As the modification of the PDC-based method, another type of pseudo-decoupling method will be improved based on elliptical parameter compensation (EPC) for much higher tracking precision, an amplification coefficient and a coupling coefficient will be introduced to mathematically construct the EPC-based model. A series of FEA simulations will also be conducted on a conventional 2-DOF PDCM to calculate the amplification and coupling coefficients as well as optimize the EPC parameters under four series of elliptical parameters. The tracking precision and operational feasibility of these two new pseudo-decoupling methods on four series of elliptical trajectories will be further analyzed and discussed in detail. Meanwhile, a conventional 2-DOF PDCM will be practically adopted to build an experimental system for investigating the pseudo-decoupling performances of an EPC-based method, the input and output displacements will be measured and collected to actually calculate the amplification coefficients and coupling coefficients, further inversely solving the actual input elliptical parameters with EPC. The error distances between the expected and experimental elliptical trajectories will also be calculated and discussed. Finally, several critical conclusions on this study will be briefly summarized.

## 1. Introduction

As one of the most excellent ultra-precision machining technologies, the elliptical vibration machining (EVM) method has been broadly adopted to precisely manufacture various complex freeform and micro-structured surfaces on widely-used difficult-to-cut materials like brittle materials, ferrous materials, and carbide alloys [1,2,3], such as a Fresnel lens on optical glass and a fly-eye lens on die steel. Currently, the material removal mechanism of the EVM method has gradually been enriched and consummated more and more, thus developing various high-performance EVM devices has become one of the greatest technological bottlenecks and challenges. However, an increasing number of researchers have widely adopted many two-degree-of-freedom piezo-driven compliant mechanisms (2-DOF PDCMs) to develop non-resonant elliptical vibration machining (EVM) devices with high work frequency, high motion precision, and compact structural dimensions [4,5,6]. In theory, the EVM process must demand two high-frequency vibrations that are excited in the cutting-depth and up-feed directions, respectively, but almost all of the reported non-resonant EVM devices comprised of various popular 2-DOF PDCMs have rarely considered the adverse influences of inherent cross-coupling motions or forces between two mutually perpendicular vibrations [5,6], but which will inevitably cause geometrical distortions of elliptical trajectories, further greatly deteriorating the forming accuracies of a complex micro-structured surface and optical freeform surfaces [7,8]. Therefore, it is very important to develop new types of decoupling strategies or 2-DOF PDCMs to efficiently improve the trajectory tracking precision of EVM devices.

So far, a large number of existing two degree of freedom piezo-driven compliant mechanisms (2-DOF PDCMs) that have been extensively applied in optical engineering and high-precision machining fields can potentially be employed in developing various non-resonant EVM apparatuses with expected high-performances such as micro/nano- positioning stages [9,10,11], micro/nano-manipulating grippers [12,13,14], and fast tool servo (FTS) in micro/nano-machining [15,16]. To weaken the cross-coupling motion and its adverse influences, almost all of the micro/nano- positioning XY stages with parallel 2-DOF PDCMs have generally adopted complex decoupling substructures to improve motion precision. For example, a typical kind of XY positioning stage with slight coupling motion has been constructed through skillfully designing a hybrid compliant-notch parallelogram mechanism [17], but as a series of very exact optimizations must be conducted on its complex decoupling compliant mechanism, obviously, this is not a general solution to weakening cross-coupling motions [15]. Meanwhile, almost all of the existing micro/nano-position XY stages have also widely adopted bi-symmetrical structures or serial configurations to eliminate cross-coupling motions. However, the popular bi-symmetrical structures may lead to the difficult installations of cutting tools, while the serial configurations must cause the great increments in motion inertia and the notable accumulation of motion errors [17,18,19], further sharply deteriorating their working bandwidths and motion precisions. Similarly, nearly all of the existing micro/nano-manipulators and micro/nano-grippers that have been widely applied in biologic and electronic fields also present large limitations in constructing our expected EVM devices without cross-coupling motion/force. Moreover, various reported 2-DOF FTS systems that have been applied in the micro/nano-machining field can highly precisely generate all required elliptical trajectories with arbitrary motion parameters for the EVM processes of complex micro-structured surfaces. However, there are few mature 2-DOF or multi-DOF FTS devices that have the possible potential to directly establish a desirable EVM apparatus with decoupling motions/forces, especially 2-DOF FTS systems with outstanding performances, which is due to the great difficulties in designing decoupling mechanisms as well as the great restrictions on working performances such as bandwidth.

To sum up, all reported micro/nano-positioning stages, manipulators, grippers, and 2-DOF FTS systems cannot be directly applied to construct suitable EVM devices due to the above-mentioned limitations and difficulties, so this paper introduced two novel types of pseudo-decoupling methods based on phase difference compensation (PDC) and elliptical parameter compensation (EPC) in sequence, which are expected to effectively and conveniently eliminate the geometrical distortions of elliptical trajectories generated by conventional 2-DOF PDCMs without a complex decoupling mechanism. In particular, these two PDC-based and EPC-based methods can guarantee that the EVM devices comprised of 2-DOF PDCMs strictly track our desirable elliptical trajectories with arbitrary motion parameters, but this does not mean that the intrinsic cross-coupling motions of PDCMs can be completely eliminated, thus can only be called a “pseudo-decoupling” method. Compared with our previous study on pseudo-decoupling compliant mechanisms with non-orthogonal substructures [20], these two new types of pseudo-decoupling methods can be directly applied to almost all conventional 2-DOF PDCMs without designing any extra decoupling substructures. In consideration of the involved issues in these two new pseudo-decoupling methods, the remainder of this research will be divided into several main sections as follows. In Section 2 and Section 3, the basic principles of the based-PDC and based-EPC pseudo-decoupling methods will first described in detail, respectively; finite element analyses (FEA) will be further conducted on a typical 2-DOF PDCM to investigate the influences of PDC angles and EPC values on the tracking precision of different elliptical trajectories; an amplification coefficient and a coupling coefficient will be defined to mathematically construct an EPC model. In Section 4, a series of experimental tests will be practically performed on an actual 2-DOF PDCM to demonstrate the high effectiveness and strong feasibility of these two proposed pseudo-decoupling methods in precisely tracking elliptical trajectories with different motion parameters. Finally, several critical conclusions of this study will be briefly summarized in Section 5.

## 2. Pseudo-Decoupling Method Based on Phase Difference Compensation

So far, the majority of previously developed 2-DOF PDCMs with decoupling motions have widely employed the popular bi-symmetrical structure configuration, which is orthogonally comprised of four of the same flexural beams with parallel connections and centrosymmetric configurations along two different directions. However, this kind of bi-symmetric 2-DOF PDCM may be very difficult to integrate into EVM devices due to the oversize decoupling structures and the painful cutting tool installations. Therefore, the 2-DOF PDCM configured by the monosymmetric structure only adopted right circular flexure hinges (RCFHs) and leaf spring flexure hinges (LSFHs) to construct our required EVM systems, as illustrated in Figure 1a. Unfortunately, this type of 2-DOF PDCM with non-bisymmetrical structures will inevitably cause adverse cross-coupling motions that must be removed to the best extent possible, further improving the precision of 2-DOF PDCMs in tracking elliptical trajectories. Motivated by the above reasons, we previously developed a sort of pseudo-decoupling 2-DOF PDCM that adopted a non-orthogonally decoupling configuration to improve the tracking precision of elliptical trajectories [20], as shown in Figure 1b. The innovation of the above 2-DOF PDCM lies in two decoupling LSFMs that are non-perpendicularly configured with an optimal decoupling angle (Θ ≥ 90°) instead of the perpendicular configuration (Θ = 90°) in conventional 2-DOF PDCMs.

However, the pseudo-decoupling 2-DOF PDCM shown in Figure 1b must conduct a series of intricate re-design, re-optimize, and re-construct operations on the conventional 2-DOF PDCMs shown in Figure 1a, which are not only laborious and time-consuming, but are also suitable for all elliptical trajectories with arbitrary motion parameters. Therefore, this paper will develop two novel types of pseudo-decoupling methods based on phase difference compensation (PDC) and elliptical parameter compensation (EPC), which can be directly applied to almost all conventional 2-DOF PDCMs without conducting any extra structural design and parameter optimization, thus exhibiting greater flexibility, stronger practicality, and higher efficiency.

### 2.1. Pseudo-Decoupling Principle of PDC-Based Method

As shown in Figure 2, the basic causes of bad cross-coupling motions can be distinctly revealed through investigating the kinematic principles of conventional 2-DOF PDCMs. More concretely, two input motions in the X and Y directions were exerted to 2-DOF PDCMs to generate our expected elliptical trajectories (namely the output motions in the X and Y directions), but certain differences could been clearly observed between the input motions and output motions such as scaling amplitudes and offsetting phases, as illustrated in Figure 2a,b. This is mainly due to the non-bisymmetric decoupling substructures of the 2-DOF PDCM and the inconsistent preloads of piezoelectric (PZT) actuators. As a result, these slight differences between the input and output motions will inevitably cause a certain degree of geometrical distortions on the expected elliptical trajectories involving major axes, minor axes and slanting angles, as shown in Figure 2c, further deteriorating the precision of 2-DOF PDCMs on tracking the required elliptical trajectories that have been widely demanded in the EVM processes of functional microstructured surfaces.

For a higher tracking precision of conventional 2-DOF PDCMs on different elliptical trajectories, this research first proposes a new type of pseudo-decoupling method based on phase difference compensations (PDC). Our previous studies indicated that the generated elliptical trajectories generally have more obvious distortion on slanting angles than their major and minor axes, as illustrated in Figure 2. Therefore, an angle Δ*ϕ* of phase difference compensation, namely the PDC angle, will additionally be attached to the initial phase difference angle *ϕ*_Input_ between two input motions to conveniently regulate the phase difference angle *ϕ*_Output_ between two output motions. Ultimately, there will be an optimum Δ*ϕ* that can minimize the geometrical distortion of the output elliptical trajectory, especially in controlling the slanting angle Δ*θ*, which can handily and rapidly suppress the adverse influences of cross-coupling motions and inconsistent PZT preloads on the tracking precision of elliptical trajectories, as shown in Figure 2c. Obviously, this proposed PCD-based method can be applicable in almost all of the reported traditional EVM devices.

### 2.2. Pseudo-Decoupling Optimization of PDC-Based Method

A commercial FEA software was further employed in conducting the analysis and optimization of phase difference compensation (PDC) on a traditional 2-DOF PDCM under different elliptical parameters, further investigating the influences of PDC angles on the decoupling performances of 2-DOF PDCM. Specifically, the range of PDC angles was reasonably selected as Δ*ϕ* = 0°~10°, and the input motions of 2-DOF PDCM along two different directions were selected as the sine and cosine waves with 10 μm amplitudes (*A_x_* = *A_y_* = 10 μm), respectively, whose vibration frequency *f_z_* and initial phase difference *ϕ* were taken into 1 Hz and 0°, as expressed in the following.
(1)$$\left\{ \begin{array}{l} x_{in}\left(t\right)=A_{x}\sin\left(2\pi f_{z}\cdot t\right)\\ y_{in}\left(t\right)=A_{y}\cos\left(2\pi f_{z}\cdot t+\phi +\Delta \phi \right) \end{array}\right.$$
where *x_in_* and *y_in_* represent the input motions of elliptical trajectories along the X-axis and Y-axis, respectively. *A_x_* and *A_y_* denote the amplitudes of input harmonic motions in the X-direction and Y-direction. *f_z_* stands for the vibration frequencies of elliptical trajectories, *ϕ* represents the initial phase difference between two input harmonic motions along the X-axis and Y-axis. Δ*ϕ* denotes the angle of phase difference compensation (PDC) that can ensure that the 2-DOF PDCM has the highest trajectory tracking precision.

Under six different PDC angles, the input and output elliptical trajectories generated by the 2-DOF PDCM were numerically simulated by the finite element analysis (FEA) method as well as analytically compared with the expected results, as shown in Figure 3. The amplitudes and phase difference of harmonic motions were consistently selected as *A_x_* = *A_y_* = 10 μm and *ϕ* = 0°, and the dimension parameters of the 2-DOF PDCM were selected as shown in Table 1.

As shown in Figure 3a, the output trajectory presented a very obvious shape distortion when the phase difference of two input motions did not adopt the PDC angle, namely Δ*ϕ* = 0°, and the perfect input circular trajectory was elongated along the −45° tilt direction as its output elliptical trajectory. Afterward, the elongations of the output elliptical trajectories gradually decreased with an increasing PDC angle Δ*ϕ*, and the corresponding elongation directions basically remained unchanged when the PDC angles Δ*ϕ* were less than 8.25°. The output elliptical trajectory was closest to a perfect circular shape when the PDC angle was selected as Δ*ϕ* = 8.25°, as shown in Figure 3e, which means that an optimum Δ*ϕ* can guarantee that this 2-DOF PDCM has the highest trajectory tracking precision. However, the trajectory tracking precision may be deteriorated further when the PDC angles Δ*ϕ* are more than 8.25°, as shown in Figure 3e, where the output elliptical trajectory was elongated along the direction with a 45° tilt angle when PDC angle was taken as Δ*ϕ* = 10°. To quantitatively evaluate the relationships between the tracking precision and PDC angles, the least square method (LSM) was further employed in mathematically fitting the above FEA-based elliptical trajectories and calculating their geometric parameters, which involved slanting angles, major axes, and minor axes. Meanwhile, a dimensionless aspect ratio *λ* was defined into the relative ratio of major axis *a* to minor axis *b*, namely *λ* = *a*/*b*, which can exactly and effectively characterize the shape distortions and attitude rotation of the output elliptical trajectories under different PDC angles. For quantitative analysis, the arithmetic average error *d_av_* and root mean square error *d_rm_* of the absolute distances between the perfect and actual output elliptical trajectories must be mathematically defined by Equation (2), respectively; all obtained results are listed in Table 2.
(2)$$\{\begin{array}{cc} \begin{array}{l} d_{i}=\sqrt{\left(x_{i}^{\mathrm{perfect}}-x_{i}^{\mathrm{actual}}\right)+\left(y_{i}^{\mathrm{perfect}}-y_{i}^{\mathrm{actual}}\right)}\\ \begin{array}{cc} d_{av}=\sum_{i=1}^{N}d_{i}/N; & d_{rm}=\sqrt{\sum_{i=1}^{N}d_{i}^{2}/N} \end{array} \end{array} & \begin{array}{cc} ; & i=1,2\cdots N. \end{array} \end{array}$$
where [*x_i_*^perfect^, *y_i_*^perfect^] and [*x_i_*^actual^, *y_i_*^actual^] denote the coordinate positions of the *i*-th data point on the perfectly expected and actually simulated/measured elliptical trajectories, respectively; *d_av_* and *d_rm_* stand for the arithmetic average and root mean square values of absolute distances between perfect and actual elliptical trajectories; *N* is the total number of data points on an elliptical trajectory.

Based on the fitted geometrical parameters of the FEA-simulated elliptical trajectories shown in Figure 3, here, we further reveal the influences of PDC angles Δ*ϕ* on the major axes *a*, minor axes *b*, and their relative aspect ratios *λ*, as shown in Figure 4. The semi-major axes *a* of the FEA-based elliptical trajectories will gradually decrease with increasing PDC angle (Δ*ϕ* < 8.25°), which will slightly increase with increasing PDC angle (Δ*ϕ* > 8.25°); as shown in Figure 4a, the semi-minor axes *b* of the output elliptical trajectories clearly exhibited an opposite changing regulation with the above semi-major axes *a*. The 2-DOF PDCM will have the highest tracking precision when the PDC angle is selected as Δ*ϕ* = 8.25°, the corresponding axis *a* and semi-minor axis *b* are 10.329 μm and 10.196 μm, respectively, which are very close to the major-axis and minor-axis lengths of perfect circular trajectories (namely *a* = *b* = 10 μm). The relative aspect ratio *λ* has a minimum value (*λ* = 1.013) when the PDC angle is selected as Δ*ϕ* = 8.25°, which is also very close to perfect *λ* = 1.0. At this moment, the error distances (*d_av_* = 0.2627 μm and *d_rm_* = 0.2676 μm) are much less than the error distances of the PDC angle Δ*ϕ* = 0° (*d_av_*= 0.5706 μm and *d_rm_* = 0.6747 μm), which cannot be completely eliminated with this proposed PDC-based method; this is due to a slight degree in the trajectory magnification of 2-DOF PDCM, as shown in Figure 4b. However, all of the obtained results clearly indicate that this proposed PDC-based method can largely improve the precision of conventional 2-DOF PDCMs on tracking the expected circular trajectories with the same motion parameters.

However, the above investigations on phase difference compensation (PDC) were only conducted on a special circular trajectory with the same amplitude and phase difference; the motion parameters of elliptical trajectories must be practically adjusted in different EVM processes. Therefore, the primary influences of PDC angles on the tracking precision of elliptical trajectories with different motion parameters must be further investigated with similar FEA methods; their corresponding output elliptical trajectories and optimum PDC angles were respectively simulated and compared in detail, as illustrated in Figure 5. Through fitting the FEA-simulated elliptical trajectories with different input motion parameters, this section quantitatively reveals the influences of initial phase difference *ϕ*, X-axis amplitude *A_x_*, Y-axis amplitude *A_y_*, and PDC angle Δ*ϕ* on the trajectory tracking precision, involving the semi-major axis *a*, semi-minor axis *b*, aspect ratio *λ*, slanting angle *θ*, arithmetic average error *d_av_*, and root mean square error *d_rm_*, as listed in Table 3.

In general, the initial phase difference *ϕ* will have a strong influence on the optimal PDC angle Δ*ϕ* when the motion amplitudes along the X-axis and Y-axis are equal (*A_x_* = *A_y_* = 10 μm), as shown in Figure 5a,b,e,f. In particular, the initial phase difference *ϕ* may exhibit a slight influence on the optimal PDC angle Δ*ϕ*, even when the two-direction amplitudes are the same, as illustrated in Figure 3e and Figure 5c,d, where the corresponding relative errors between the perfect and actual aspect ratios *λ* were 1.30%, 5.72%, and 3.17%. As listed in Table 3, most relative aspect ratios *λ* are very close to their perfect values, which well demonstrates that an optimal PDC angle can effectively improve the precision of the conventional 2-DOF PDCM on tracking the expected elliptical trajectories under different motion parameters. In addition, we can distinctly know that the amplitudes in two directions may have different degrees of influence on the optimal PDC angle Δ*ϕ* when the initial phase difference *ϕ* is unchanged, as shown in Figure 3e and Figure 5a,b. For example, when the amplitudes in two directions are consistently selected as *A_x_* = 10 μm and *A_y_* = 5 μm, the optimal PDC angles Δ*ϕ* will be respectively determined as 10.5° and 15.0° when the initial phase differences *ϕ* are set as 0° and 30°, as shown in Figure 5a,e.

Together with the expected output elliptical trajectories, the actual output elliptical trajectories may be slightly amplified when the initial phase differences *ϕ* are chosen as positive agrees, as shown in Figure 5d,f, but which will be lightly shrunken when the initial phase differences *ϕ* are set as positive agrees, as shown in Figure 5c,e. When the initial phase differences are selected as *ϕ* = 0°, the magnification or reduction in elliptical trajectories can be neglected, as shown in Figure 5a,b. Specifically, the respective errors between the perfect and actual aspect ratio *λ* were 0.99% and 0.84% when the initial phase differences were selected as *ϕ* = 0°, which were much lower than the aspect ratio errors when the initial phase differences were selected as *ϕ* ≠ 0°, and the maximum error reached 16.96% when *ϕ* = 30°. Similarly, the arithmetic average errors *d_av_* and root mean square errors *d_rm_* were respectively less than 0.19 μm and 0.20 μm when *ϕ* = 0°, which were also much lower than the error distances when *ϕ* ≠ 0°, and the maximum *d_av_* and *d_rm_* were more than 0.86 μm and 0.87 μm when *ϕ* = −45°. In short, the optimal PDC angles will have different degrees of dependence on the input motion parameters. Conversely, the input motion parameters will have some degree of influence on the tracking precision of elliptical trajectories. However, the above obtained intricate relationships will increase the application difficulty and decrease the tracking precision of this proposed PDC-based pseudo-decoupling method.

In summary, the cross-coupling motions of various conventional 2-DOF PDCMs will inevitably cause obvious tracking errors such as attitude rotations and shape distortions on elliptical trajectories that have been widely demanded in EVM processes. Thus, a novel type of PDC-based pseudo-decoupling method was proposed to improve the precision of conventional 2-DOF PDCMs on tracking elliptical trajectories. Unfortunately, the critical PDC angles Δ*ϕ* depend largely on the input motion parameters of elliptical trajectories, involving initial phase differences *ϕ* and two-direction amplitudes (*A_x_* and *A_y_*). Therefore, if we want to determine the optimum PDC angles Δ*ϕ* that can efficiently guarantee that all 2-DOF PDCMs have excellent tracking precision on the expected elliptical trajectories with different motion parameters, both the initial phase differences and two-direction amplitudes must be further taken into consideration.

## 3. Pseudo-Decoupling Method Based on Elliptical Parameter Compensations

From the above PDC-based pseudo-decoupling analyses on conventional 2-DOF PDCMs, the slanting angles *θ* of different output elliptical trajectories can be effectively modified through optimizing the PDC angles, but this type of PDC-based pseudo-decoupling method still exhibits three main aspects of deficiencies: (a) The output elliptical trajectories after modification by the PDC-based method can only ensure near planar attitudes (namely slanting angle *θ*) with perfect input elliptical trajectories under different motion parameters, but the semi-major axes *a*, semi-minor axes *b*, and aspect ratio λ still show obvious geometrical deviations; (b) the optimization of PDC angles Δ*ϕ* only takes the initial phase differences *ϕ* into consideration, but the two-direction amplitudes (*A_x_* and *A_y_*) also exhibit significant influence on the optimum PDC angles Δ*ϕ*, thus the pseudo-decoupling method with higher accuracy must synchronously consider the initial phase differences *ϕ* and two-direction amplitudes; (c) a series of laborious FEA simulations need to be repeatedly conducted on conventional 2-DOF PDCMs under different elliptical trajectories, so PDC-based pseudo-decoupling analyses and optimization are time-consuming and labor-intensive. As an improvement, this study will further develop another type of pseudo-decoupling method based on elliptical parameter compensation (EPC), namely the EPC-based method, which can obtain a higher tracking precision than the PDC-based method through taking both the initial phase difference *ϕ* and two-direction amplitudes (*A_x_* and *A_y_*) into consideration. Meanwhile, an amplification coefficient and a coupling coefficient will be introduced to mathematically describe the rotating attitudes and scaling dimensions between the input and output elliptical trajectories of conventional 2-DOF PDCMs.

### 3.1. Basic Principle of Elliptical Parameter Compensations

To exactly generate the expected elliptical trajectories, two typical harmonic motions expressed by Equation (1) will be respectively exerted on the input ends of the 2-DOF PDCM along its X-direction and Y-direction. Afterward, taking both the motion amplification coefficient *κ* and motion coupling coefficient *ω* into account, the actual output elliptical trajectory of the adopted 2-DOF PDCM can be mathematically formulated by the following:(3)$$\left\{ \begin{array}{l} x_{out}\left(t\right)=\kappa x_{in}\left(t\right)+\kappa \omega y_{in}\left(t\right)=\kappa A_{x}\sin(2\pi f_{z}\cdot t)+\kappa \omega A_{y}\cos\left(2\pi f_{z}\cdot t+\phi +\Delta \phi \right)\\ y_{out}\left(t\right)=\kappa y_{in}\left(t\right)+\kappa \omega x_{in}\left(t\right)=\kappa A_{y}\cos(2\pi f_{z}\cdot t+\phi +\Delta \phi )+\kappa \omega A_{x}\sin\left(2\pi f_{z}\cdot t\right) \end{array}\right.$$
where *x_out_*(*t*) and *y_out_*(*t*) denote the output motion trajectories of the 2-DOF PDCM along the X-direction and Y-direction under the input motion functions *x_in_*(*t*) and *y_in_*(*t*), respectively; *κ* and *ω* stand for the newly introduced motion amplification coefficient and coupling coefficient between the input and output elliptical trajectories, respectively. With the difference formulas of trigonometric functions, the above output elliptical trajectory formulations in Equation (3) can be further deduced by the below:(4)$$\left\{ \begin{array}{l} x_{out}\left(t\right)=\kappa A_{x}\sin\left(2\pi f_{z}\cdot t\right)+\kappa \omega A_{y}\cos\left(2\pi f_{z}\cdot t\right)\cos\left(\phi +\Delta \phi \right)-\kappa \omega A_{y}\sin\left(2\pi f_{z}\cdot t\right)\sin\left(\phi +\Delta \phi \right)\\ y_{out}\left(t\right)=\kappa A_{y}\cos\left(2\pi f_{z}\cdot t\right)\cos\left(\phi +\Delta \phi \right)-\kappa A_{y}\sin\left(2\pi f_{z}\cdot t\right)\sin\left(\phi +\Delta \phi \right)+\kappa \omega A_{x}\sin\left(2\pi f_{z}\cdot t\right) \end{array}\right.$$
(5)$$\left\{ \begin{array}{l} x_{out}\left(t\right)=\left[\kappa A_{x}-\kappa \omega A_{y}\sin\left(\phi +\Delta \phi \right)\right]\sin\left(2\pi f_{z}\cdot t\right)+\kappa \omega A_{y}\cos\left(\phi +\Delta \phi \right)\cos\left(2\pi f_{z}\cdot t\right)\\ y_{out}\left(t\right)=\left[\kappa \omega A_{x}-\kappa A_{y}\sin\left(\phi +\Delta \phi \right)\right]\sin\left(2\pi f_{z}\cdot t\right)+\kappa A_{y}\cos\left(\phi +\Delta \phi \right)\cos\left(2\pi f_{z}\cdot t\right) \end{array}\right.$$

Afterward, the auxiliary angle formulas of trigonometric functions are respectively introduced to the above output motion formulations and concisely derived as follows:(6)$$\left\{ \begin{array}{l} x_{out}\left(t\right)=A_{x0}\sin\left(2\pi f_{z}\cdot t\right)=\sqrt{a_{x}{}^{2}+b_{x}{}^{2}}\sin\left(2\pi f_{z}\cdot t+\phi_{x}\right)\\ y_{out}\left(t\right)=A_{y0}\cos\left(2\pi f_{z}\cdot t+\phi_{0}\right)=\sqrt{a_{y}{}^{2}+b_{y}{}^{2}}\cos\left(2\pi f_{z}\cdot t-\phi_{y}\right) \end{array}\right.$$
where *ϕ*_0_ and *A_x_*_0_, *A_y_*_0_ denote the initial phase differences, X-direction amplitudes, and Y-direction amplitudes of the output elliptical trajectories, respectively, which are directly related to some intermediate variables *a_x_*, *b_x_*, *ϕ_x_* and *a_y_*, *b_y_*, *ϕ_y_*; their corresponding mathematical formulas can be specifically expressed by the below:(7)$$\left\{ \begin{array}{l} \begin{array}{ccc} a_{x}=\kappa A_{x}-\kappa \omega A_{y}\sin\left(\phi +\Delta \phi \right); & b_{x}=\kappa \omega A_{y}\cos\left(\phi +\Delta \phi \right); & \phi_{x}=\arctan b_{x}/a_{x} \end{array}\\ a_{y}=\begin{array}{ccc} \kappa \omega A_{x}-\kappa A_{y}\sin\left(\phi +\Delta \phi \right); & b_{y}=\kappa A_{y}\cos\left(\phi +\Delta \phi \right); & \phi_{y}=\arctan a_{y}/b_{y} \end{array} \end{array}\right.$$
(8)$$\left\{ \begin{array}{l} A_{x0}=\sqrt{a_{x}^{2}+b_{x}^{2}}=\kappa \sqrt{{\left(A_{x}\right)}^{2}-2\omega A_{x}A_{y}\sin\left(\phi +\Delta \phi \right)+{\left(\omega A_{y}\right)}^{2}}\\ A_{y0}=\sqrt{a_{y}^{2}+b_{y}^{2}}=\kappa \sqrt{{\left(\omega A_{x}\right)}^{2}-2\omega A_{x}A_{y}\sin\left(\phi +\Delta \phi \right)+{\left(A_{y}\right)}^{2}}\\ \phi_{0}=\arctan\frac{A_{y}\cos\left(\phi +\Delta \phi \right)}{\omega A_{x}-A_{y}\sin\left(\phi +\Delta \phi \right)}-\arctan\frac{\omega A_{y}\cos\left(\phi +\Delta \phi \right)}{A_{x}-\omega A_{y}\sin\left(\phi +\Delta \phi \right)} \end{array}\right.$$

Ultimately, this type of EPC-based pseudo-decoupling method is mathematically constructed in extreme detail, and its mathematical model can simultaneously compensate three motion parameters of input elliptical trajectories involving *A_x_*, *A_y_*, and *ϕ*, thus strictly ensuring that 2-DOF PDCM can more accurately track the expected output elliptical trajectories whose actual output motion parameters *A_x_*_0_, *A_y_*_0_, and *ϕ*_0_ are expressed by Equation (8).

### 3.2. Optimization of Amplification and Coupling Coefficients

According to the above constructed EPC mathematical model, this paper can distinctly determine the main relationships between the input elliptical trajectories (whose parameters are *A_x_*, *A_y_*, and *ϕ*) and output elliptical trajectories (whose parameters are *A_x_*_0_, *A_y_*_0_, and *ϕ*_0_), but it is very important to effectively and precisely determine the defined amplification coefficients *κ* and coupling coefficients *ω* of conventional 2-DOF PDCMs. In general, the matrix-based compliant model of 2-DOF PDCM needs to be theoretically constructed to calculate its amplification coefficient *κ* and coupling coefficients *ω*, but it is very difficult and tortuous. Here, we will directly employ the cost-effective FEA method in numerically revealing the relationships between the input and output elliptical trajectories as well as the key influences of the amplification coefficient and coupling coefficient on tracking precision. Afterward, the FEA-simulated results will be introduced into the above EPC-based model for calculating the motion amplification coefficient and coupling coefficient of a conventional 2-DOF PDCM whose size parameters were chosen as those in Table 1. To optimize amplification coefficient *κ* and coupling coefficient *ω* of the adopted 2-DOF PDCM, their key influences on the arithmetic average deviation *ξ_av_* and root mean squared deviation *ξ_rm_* of the absolute distances between the EPC-calculated and FEA-simulated elliptical trajectory were analytically investigated, as illustrated in Figure 6.
(9)$$\{\begin{array}{cc} \begin{array}{l} \xi_{i}=\sqrt{\left(x_{i}^{\mathrm{Cal}}-x_{i}^{\mathrm{Sim}}\right)+\left(y_{i}^{\mathrm{Cal}}-y_{i}^{\mathrm{Sim}}\right)}\\ \begin{array}{cc} \xi_{av}=\sum_{i=1}^{N}\xi_{i}/N; & \xi_{rm}=\sqrt{\sum_{i=1}^{N}\xi_{i}^{2}/N} \end{array} \end{array} & ;i=1,2\cdots N \end{array}$$
where [*x_i_*^Cal^, *y_i_*^Cal^] and [*x_i_*^Sim^, *y_i_*^Sim^] denote the coordinate positions of the *i*-th data point on the elliptical trajectories obtained by the EPC-based calculations and FEA-based simulations, respectively; *N* is the total number of data points on an elliptical trajectory.

As shown in Figure 6, both the arithmetic average deviation *ξ_av_* and root mean squared deviation *ξ_rm_* exhibited a non-monotonic correlation with motion amplification coefficient *κ* and coupling coefficient *ω*, but the amplification coefficient *κ* presented a stronger influence on error deviation *ξ_av_* and *ξ_rm_* than the coupling coefficient *ω*, which distinctly indicates that this actually adopted 2-DOF PDCM had the optimum amplification coefficient and coupling coefficient (*κ* = 1.0328 and *ω* = −0.0732); corresponding error deviations were *ξ_av_* = 2.20 × 10^−4^ μm and *ξ_rm_*= 2.60 × 10^−4^ μm. However, the amplification coefficient *κ* and coupling coefficient *ω* both had relatively low values, which was because the structural dimensions of this adopted 2-DOF PDCM were strictly optimized in previous research [20]. It is very important to note that the coupling coefficient *ω* is a small negative value, which means that an input actuated motion along the positive X-direction will slightly cause an output cross-coupling motion along the negative Y-direction, and vice versa. With four series of elliptical trajectories, the key influences of the input motion parameters on the error distances *d_av_* and *d_rm_* between the calculated (abbreviated as Cal) and expected (abbreviated as Exp) trajectories as well as the error deviations *ξ_av_* and *ξ_rm_* between the calculated and simulated (abbreviated as Sim) trajectories will be further revealed and discussed in detail, as shown in Figure 7.

For the output elliptical trajectories driven by the input motion parameters (*A_x_* = 10 μm, *A_y_* = 10 μm, *ϕ* = 0°), the error deviations between the calculated and simulated trajectories were *ξ_av_* = 2.20 × 10^−4^ μm and *ξ_rm_* = 2.60 × 10^−4^ μm, as shown in Figure 7a. Another two output elliptical trajectories generated by the input motion parameters (*A_x_* = 10 μm, *A_y_* = 5 μm, *ϕ* = 0° and *A_x_* = 5 μm, *A_y_* = 10 μm, *ϕ* = 45°) also had very high calculation accuracies, whose error deviations were *ξ_av_* = 1.72 × 10^−4^ μm, *ξ_rm_* = 2.22 × 10^−4^ μm and *ξ_av_* = 2.12 × 10^−4^ μm, *ξ_rm_* = 2.55 × 10^−4^ μm, as illustrated in Figure 7b,d. However, the output elliptical trajectories generated by the input motion parameters (*A_x_* = 10 μm, *A_y_* = 10 μm, *ϕ* = 45°) had slightly higher deviations (*ξ_av_* = 3.59 × 10^−4^ μm and *ξ_rm_* = 4.61 × 10^−4^ μm) than the former three series of output elliptical trajectories, but which were still much less than the major-axis or minor-axis length of the output elliptical trajectories. Furthermore, the resulted arithmetic average distances *d_av_* and root mean squared distances *d_rm_* between the calculated and expected elliptical trajectories clearly indicate that the input motion parameters have great influences on their tracking precision, the output elliptical trajectory (*A_x_* = 5 μm, *A_y_* = 10 μm, *ϕ* = 45°) had the worst tracking precision (*d_av_* = 0.6116 μm and *d_rm_* = 0.7532 μm), and the output elliptical trajectory (*A_x_* = 10 μm, *A_y_* = 5 μm, *ϕ* = 0°) had the highest tracking precision (*d_av_* = 0.5037 μm and *d_rm_* = 0.5954 μm), but their peak-to-valley (PV) values of the four elliptical trajectories were greater than 1 μm. In summary, a very good agreement could clearly be found between the theoretical calculations and FEA-based simulations under different elliptical trajectories, which can indicate that the amplification coefficient *κ* and coupling coefficient *ω* of the 2-DOF PDCM will mainly depend on its structure parameters rather than the input motion parameters of the elliptical trajectories. However, all of obtained results shown in Figure 7 strongly demonstrate the effectiveness and feasibility of the built mathematical model in exactly describing the intricate relationship between the input motion parameters and output elliptical trajectories of conventional 2-DOF PDCMs.

### 3.3. Verification of Elliptical Parameter Compensations

The theoretically deduced EPC-based model can precisely forecast output elliptical trajectories under four series of different input motion parameters, but which is only the foundation of conducting the elliptical parameter compensation (EPC). This EPC-based pseudo-decoupling model will be further utilized to reverse calculate the optimal EPC values (namely *A_x_*, *A_y_*, and *ϕ*) according to the expected output elliptical trajectories (namely *A_x_*_0_, *A_y_*_0_, and *ϕ*_0_). Ultimately, the compensated two-direction amplitudes *A_x_*, *A_y_* and initial phase difference *ϕ* will ensure that the output elliptical trajectories can coincide with all of the expected elliptical trajectories as much as possible. Under four series of expected output elliptical trajectories, the actual input motion parameters modified by the EPC inverse model were exerted on the two-direction input ends of the 2-DOF PDCM to generate elliptical trajectories through FEA-based simulations and EPC-based calculations, respectively, and their error distances related to the expected trajectories were calculated to quantitatively validate the effectiveness and feasibility of the EPC-based method, as shown in Figure 8. Meanwhile, the EPC optimums of the actual input motion parameters, the arithmetical average distances *d_av_*^1^ and root mean squared distances *d_rm_*^1^ between the FEA-simulated and expected output elliptical trajectories, and the arithmetical average distances *d_av_*^2^ and root mean squared *d_rm_*^2^ between the EPC-calculated and expected output elliptical trajectories need to be further investigated and fairly compared in more detail, respectively, as listed in Table 4.

To exactly generate our expected output circular trajectory (*A_x_*_0_ = *A_y_*_0_ = 10 μm, *ϕ*_0_ = 0°), the inverse solution of the EPC-based model was first conducted to calculate the actual input motion parameters (*A_x_* = *A_y_* = 9.7606 μm and *ϕ* = −8.3732°), which were then practically introduced into the FEA-based simulation and EPC-based model to generate the expected output circular trajectory, as illustrated in Figure 8a, where its corresponding error distances were *d_av_*^1^ = 2.4589 × 10^−4^ μm, *d_rm_*^1^ = 2.4848 × 10^−4^ μm, and *d_av_*^2^ = 2.3139 × 10^−5^ μm, *d_rm_*^2^ = 2.3224 × 10^−5^ μm, which were mainly caused by solution errors of the FEA and mathematical software, and both were much lower than the radius of circular trajectory, so these can be completely ignored.

However, the error distances *d_av_*^1^ and *d_rm_*^1^ of the FEA-based results were apparently higher than the *d_av_*^2^ and *d_rm_*^2^ of the EPC-based trajectory, which appeared as very obvious and sharp fluctuations; this may be due to the inherent calculation and theoretical errors in FEA computation. Similarly, another three series of output elliptical trajectories with different motion parameters were also investigated and discussed in detail, respectively, as shown in Figure 8b–d, where very similar relationships and conclusions can also be distinctly revealed and summarized. There were positive correlations among the resulted error distances in the FEA-simulated and EPC-calculated elliptical trajectories shown in Figure 8a,c,d, while the error distances of the FEA-simulations had negative correlations with the EPC-based calculations, as illustrated in Figure 8b. In theory, the error distances of circular trajectory shown in Figure 8a had an optimum peak-to-valley (PV) value (about 0.554 × 10^−5^ μm), but the PV values of error distances increased to 2.485 × 10^−5^ μm, 2.90 × 10^−5^ μm, and 1.366 × 10^−5^ μm when the input motion parameters were altered. From all of the resulting output elliptical trajectories shown in Figure 8, the initial phase differences generally possessed stronger deteriorations on the PV values of the error distances than the two-direction amplitudes, but both the amplitudes and phase differences had a slight influence on the error distances *d_av_* and *d_rm_*.

In addition, all of the obtained results illustrated in Figure 8 and Table 4 clearly indicate that these actual input elliptical parameters modified by the EPC-based pseudo-decoupling model can effectively improve the tracking precision of the 2-DOF PDCM on our expected elliptical trajectories. Meanwhile, the output elliptical trajectories obtained by the FEA-based simulations and EPC-based calculations were very approximate to the expected trajectories, and the corresponding error distances *d_av_* and *d_rm_* were much lower than those of the initial PDC-based method listed in Table 1. However, all of the resulted error distances *d_av_* and *d_rm_* were much lower than the lengths of the minor-axes or major semi-axes of the elliptical trajectories, so can also be neglected and well demonstrate that the high effectiveness and strong feasibility of this proposed EPC-based pseudo-decoupling method in precisely tracking required elliptical trajectories in various EVM processes.

## 4. Experiments and Discussion

The effectiveness and feasibility of the proposed EPC-based pseudo-decoupling method were finely demonstrated by conducting FEA simulations and theoretical calculations on a conventional 2-DOF PDCM, but still needs to be validated with actual experiments, as shown in Figure 9. First, two piezoelectric stack (PZT) actuators (PI P−887.51, nominal travel range is 15 μm) were mounted in two input-ends of a 2-DOF PDCM manufactured by wire-electrode cutting; two low-frequency (1 Hz) harmonic signals were generated by a programmable multi-axes controller (Delta Tau PMAC, Kyoto, Japan) to drive these two PZT actuators. Subsequently, the real-time input and output displacements in two different directions were practically measured and collected by four capacitive probes (MicroSense 2805, Lowell, MA, USA) and a multi-channel position measuring module (MicroSense II Model 5300, Lowell, MA, USA); their measuring scope and resolution is ±100 μm and 1 nm. All experimental tests were performed on an air-bearing vibration-isolated platform (Newport RS4000, Irvine, CA, USA) to avoid the influences of disturbances and noise from the external environment, and the channel noise of four capacitive displacement sensors were respectively measured and collected, as shown in Figure 10a, where the peak-to-valley (PV) values of four channel noises were very similar and less than 25 nm, so there will have a very light influence on the experimental tests.

In light of the manufacturing imperfections of PDCM and the inconsistent preloads of PZT actuators, this actual PDCM system has certain differences from its FEA model. Therefore, four series of expected elliptical trajectories were first adopted to excite the PDCM, as shown in Figure 10b, then the input and output displacements along the X and Y directions were respectively collected to solve the actual amplification coefficient *κ* and coupling coefficient *ω*, as shown in Figure 10. Unlike the constant coefficients *κ* and *ω* obtained in the FEA simulation shown in Figure 6, the experimental results clearly showed that the elliptical parameters also had obvious influences on the actual coefficients *κ* and *ω*, which was mainly due to the driving forces of the PZT actuators undergoing intricate variations with increasing input displacements; in other words, the input stiffnesses of the 2-DOF PDCM will have nonlinear relationships with its input displacements along the X and Y directions. Furthermore, all elliptical trajectories must be strictly limited to the range of 0~10 μm rather than −10~10 μm, which is due to all PZT actuators not being able to generate negative displacement under negative voltage, otherwise it may cause damage to the PZT actuators, as shown in Figure 10b.

Next, the input and output displacements without EPC modification were firstly measured and collected to solve the actual coefficients *κ* and *ω* under different motion parameters, their optimum values were selected based on the relation surfaces shown in Figure 11. Then the calculated coefficients *κ* and *ω* were further employed in calculating our required input elliptical parameters with EPC modification, which ensured that the output elliptical trajectories were very close to our expected trajectories. Under four series of different elliptical trajectories, the actual input elliptical parameters with EPC, amplification coefficient *κ*, and coupling coefficient *ω* as well as the error distances between the expected and experimental elliptical trajectories were respectively calculated and listed in Table 5; corresponding elliptical trajectories were also compared and discussed in detail, as shown in Figure 12. 

As listed in Table 5, the established 2-DOF PDCM system could output a nearly perfect elliptical trajectory (*A_x_*_0_ = *A_y_*_0_ = 5.0 μm, *ϕ*_0_ = 0°) when the actual input elliptical parameters were modified by the EPC method, namely *A_x_* = *A_y_* = 5.0311 μm and *ϕ* = −6.1248°. At this moment, the amplification coefficient and coupling coefficient were *ω* = −0.0535 and *κ* = 0.9981; both had obvious differences from the FEA-simulation results shown in Figure 6. The average value and RMS value of the error distances between the expected and experimental elliptical trajectories were about *d_av_* = 0.0658 μm and *d_rm_* = 0.0756 μm, respectively, which clearly shows that the experimental output elliptical trajectory after EPC modification was very close to the expected elliptical trajectory, exhibiting a good agreement with the trajectory comparison illustrated in Figure 12a. Similarly, two series of expected elliptical trajectories (*A_x_*_0_ = 5.0 μm, *A_y_*_0_ = 2.5 μm, *ϕ*_0_ = 0° and *A_x_*_0_ = 2.5 μm, *A_y_*_0_ = 5.0 μm, *ϕ*_0_ = 45°) exhibited very high tracking precision between the expected and experimental results; the average distances and RMS distances were about *d_av_* = 0.0286 μm, *d_rm_* = 0.0360 μm and *d_av_* = 0.0523 μm, *d_rm_* = 0.0634 μm, respectively, which were much lower than the cross-coupling motions of the 2-DOF PDCM before EPC modification, thus these error distances can be completely acceptable when we take the channel noise levels (about 0.025 μm) of the experimental measuring system into account. The corresponding output elliptical trajectories are illustrated in Figure 12b,d.

As shown in Figure 12c, for the expected output elliptical trajectory with *A_x_*_0_ = *A_y_*_0_ = 5.0 μm and *ϕ*_0_ = 45°, the average and RMS values of the error distances between the expected and experimental results were *d_av_* = 0.0707 μm and *d_rm_* = 0.0869 μm, respectively, which were slightly greater than the former three series of elliptical trajectories; this is due mainly to the actual amplification coefficient *κ* and coupling coefficient *ω*, which are mathematically calculated based on experimental input and output elliptical trajectories, that perhaps may not be accurate enough yet, as shown in Figure 11c, but the corresponding tracking precision is still acceptable, so some optimization algorithms will be employed to more precisely solve coefficients *κ* and *ω* in future work. However, all of the obtained experimental results had good agreement with the FEA-based simulation and EPC-based calculation results in the previous section, which thus well demonstrates that this proposed EPC-based method has strong feasibility and high effectiveness in eliminating the adverse cross-coupling motions that have been widely found in almost all conventional 2-DOF PDCMs, especially for parallel configuration, thus further improving the forming precision of EVM processes.

## 5. Conclusions

In order to improve the trajectory tracking precision of a widely-used EVM apparatus constructed by two-degree-of-freedom piezoelectrically driven compliant mechanisms (2-DOF PDCMs), a novel type of pseudo-decoupling method was first proposed based on phase difference compensation (PDC). As a modification of the PDC method, another type of pseudo-decoupling method was further improved based on elliptical parameter compensation (EPC), and an amplification coefficient and a coupling coefficient were defined to mathematically model the EPC-based method. Under different elliptical parameters, the FEA simulations and theoretical calculations were both conducted on a conventional 2-DOF PDCM to investigate the error distances among the FEA-simulated, EPC-calculated, and expected output elliptical trajectories. All of the obtained results strongly demonstrate that these two proposed pseudo-decoupling methods can effectively improve the trajectory tracking precision of conventional 2-DOF PDCMs. Finally, a series of actual experiments were performed to more strongly validate the feasibility and effectiveness of the EPC-based pseudo-decoupling method in exactly tracking arbitrary elliptical trajectories that have been extensively demanded in the EVM processes of functional microstructured surfaces, and several crucial conclusions from this study are briefly summarized.

(a)The developed PDC-based pseudo-decoupling method could effectively eliminate the attitude rotations of the elliptical trajectories with different motion parameters, but some degrees of dimension amplifications could obviously be observed on almost all of the elliptical trajectories. This as mainly because the decoupling substructures of 2-DOF PDCMs have a certain level of displacement magnification, so their trajectory tracking precision makes it very difficult to satisfy the practical demands in many EVM processes of high-precision functional microstructured surfaces.(b)The EPC-based pseudo-decoupling method was mathematically modeled through defining an amplification coefficient and a coupling coefficient that could effectively describe the relationships between the input motion parameters and output geometrical features of different elliptical trajectories. Under different input motion parameters, the output elliptical trajectories theoretically calculated by the EPC-based model were a very close approximation to the FEA-based simulation results, and their corresponding error distances were very slight and so can be fully negligible, which can strongly prove the high accuracy and validity of the constructed EPC model.(c)The actual input motion parameters compensated by the inverse EPC model were respectively applied on the 2-DOF PDCM to generate the expected elliptical trajectories through FEA-based simulation and EPC-based calculation. Under four series of different input motion parameters, the FEA simulated and EPC-calculated output elliptical trajectories both had very good approximation with our expected trajectories, and their corresponding error distances were very slight and could be completely ignored, which strongly prove the high effectiveness and feasibility of the EPC-based pseudo-decoupling method in improving the trajectory tracking precision of conventional 2-DOF PDCMs.(d)The input motion parameters compensated by the inverse EPC-based model were practically employed in exciting the 2-DOF PDCM of the built EVM experimental system, then the output elliptical trajectories were measured and collected. However, all of the obtained results clearly indicated the absolute error distances between the expected and experimental elliptical trajectories under four series of different motion parameters, which well demonstrated that this newly proposed EPC-based method exhibits strong feasibility and high effectiveness in eliminating the bad cross-coupling motions of conventional 2-DOF PDCMs, especially with parallel configuration.

## Figures and Tables

**Figure 1 micromachines-14-02043-f001:**
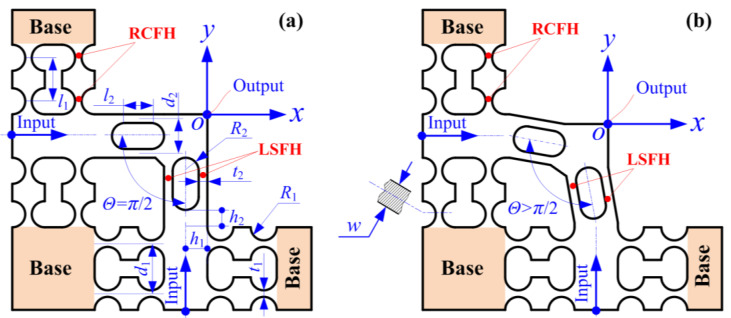
Basic principle of the developed two-DOF pseudo-decoupled compliant mechanisms. (**a**) Traditional orthogonal configuration; (**b**) novel non-orthogonal configuration [20].

**Figure 2 micromachines-14-02043-f002:**
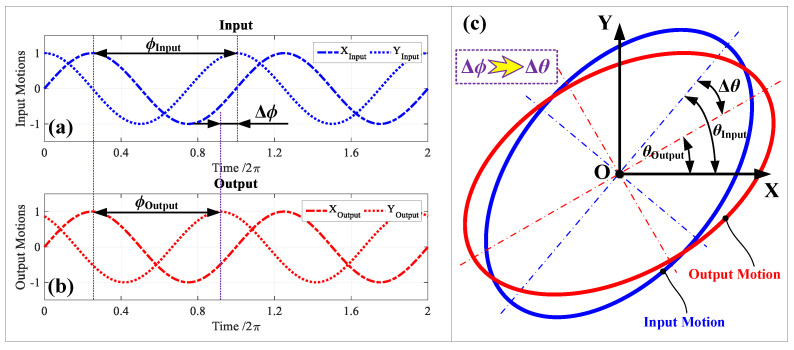
The cross-coupling motion causes and the kinematic principle of traditional 2-DOF PDCMs. (**a**) Input motions; (**b**) output motions; (**c**) cross-coupling motions.

**Figure 3 micromachines-14-02043-f003:**
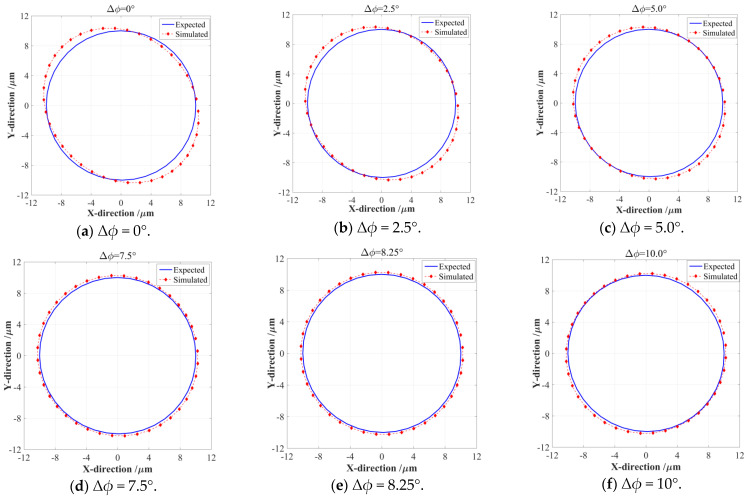
The dependence of the pseudo-decoupling performances of the 2-DOF PDCM on different PDC angles.

**Figure 4 micromachines-14-02043-f004:**
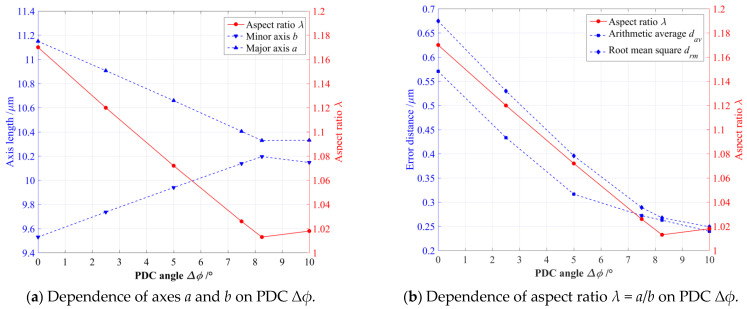
The influences of different PDC angles on (**a**) the major axes and minor axes; (**b**) their aspect ratios.

**Figure 5 micromachines-14-02043-f005:**
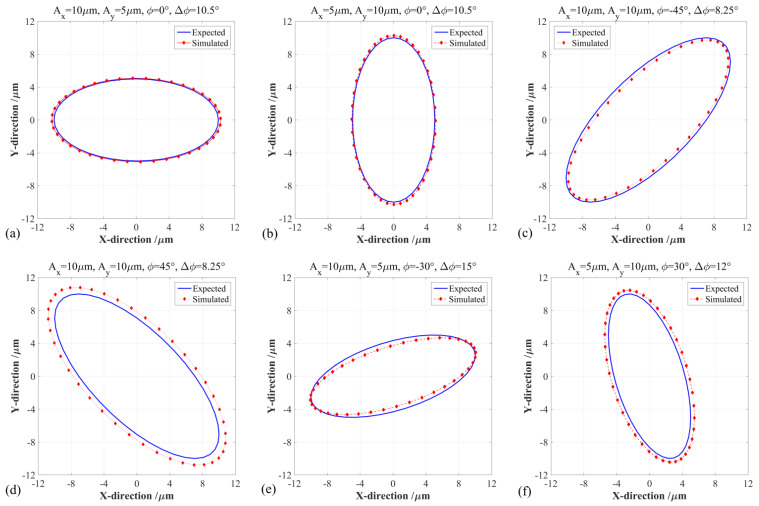
The comparisons between the input and output trajectories under different elliptical parameters and PDC angles. (**a**) *A_x_* = 10 μm, *A_y_* = 5 μm, *ϕ* = 0°, Δ*ϕ* = 10.5°; (**b**) *A_x_* = 5 μm, *A_y_* = 10 μm, *ϕ* = 0°, Δ*ϕ* = 10.5°; (**c**) *A_x_* = 10 μm, *A_y_* = 10 μm, *ϕ* = −45°, Δ*ϕ* = 8.25°; (**d**) *A_x_* = 10 μm, *A_y_* = 10 μm, *ϕ* = 45°, Δ*ϕ* = 8.25°; (**e**) *A_x_* = 5 μm, *A_y_* = 10 μm, *ϕ* = −30°, Δ*ϕ* = 15.0°; (**f**) *A_x_* = 10 μm, *A_y_* = 5 μm, *ϕ* = 30°, Δ*ϕ* = 12.0°.

**Figure 6 micromachines-14-02043-f006:**
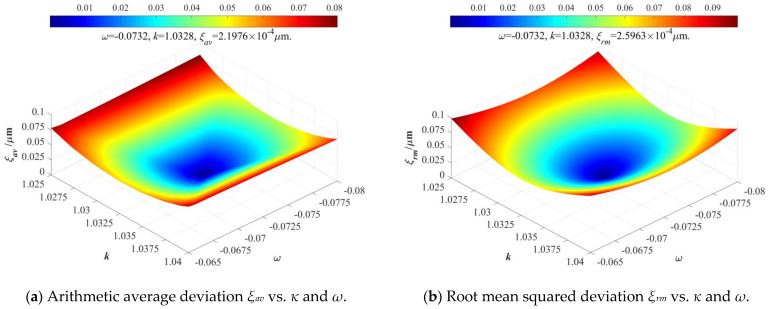
The dependences of distance deviations on the amplification coefficient *κ* and coupling coefficient *ω*.

**Figure 7 micromachines-14-02043-f007:**
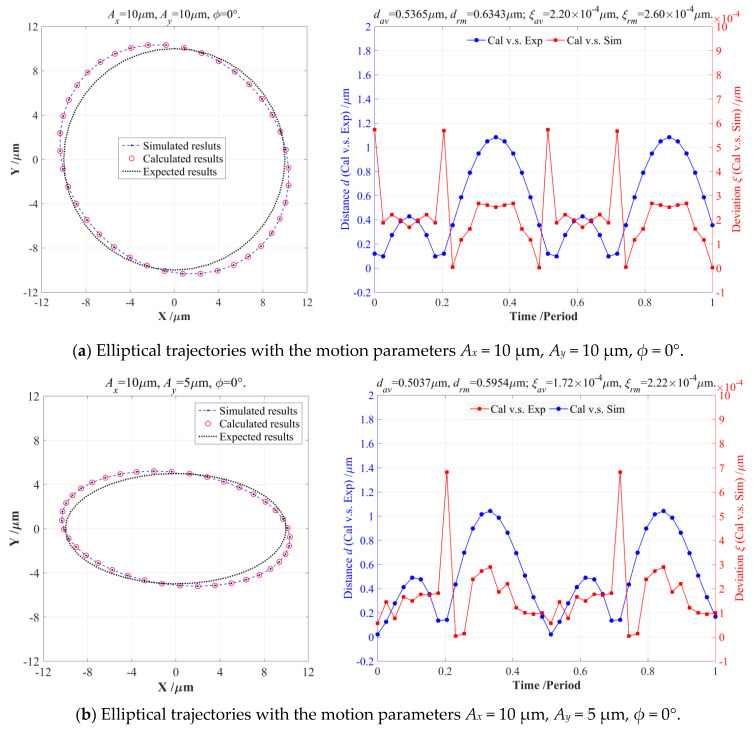
Comparisons of the FEA-simulations and EPC-calculation under different output elliptical trajectories.

**Figure 8 micromachines-14-02043-f008:**
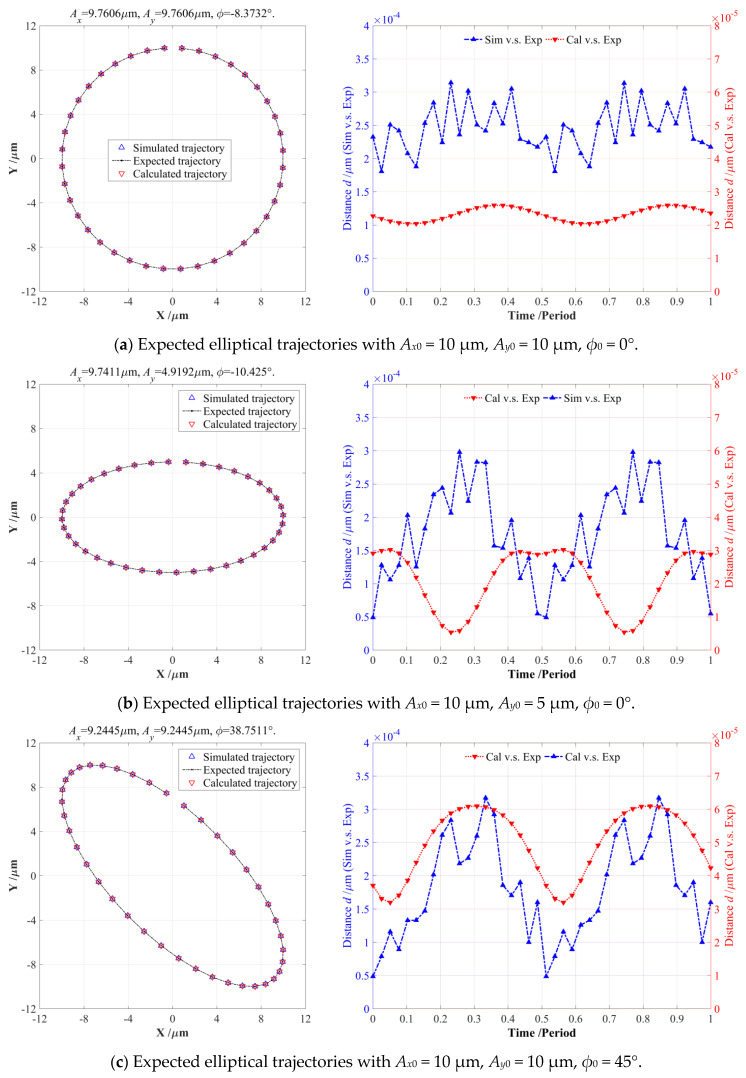
Comparison of the FEA-simulated and EPC-calculated elliptical trajectories with different motion parameters.

**Figure 9 micromachines-14-02043-f009:**
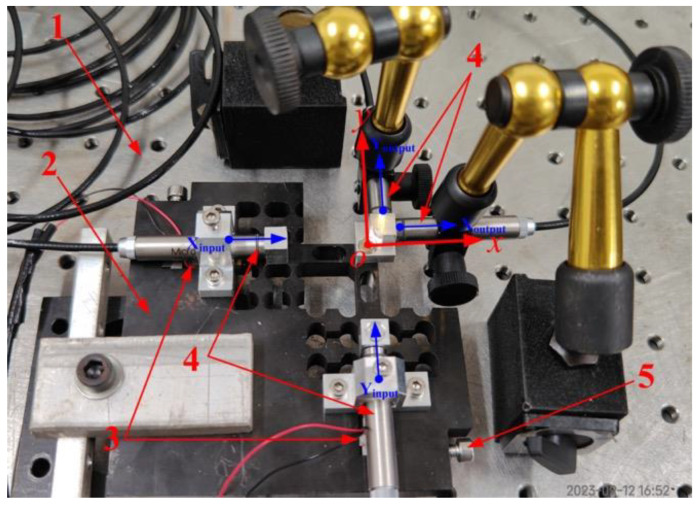
Experiment setup. 1—Vibration-isolated platform; 2—Compliant mechanism; 3—Piezoelectric actuators; 4—Capacitive displacement sensors; 5—Preload screws.

**Figure 10 micromachines-14-02043-f010:**
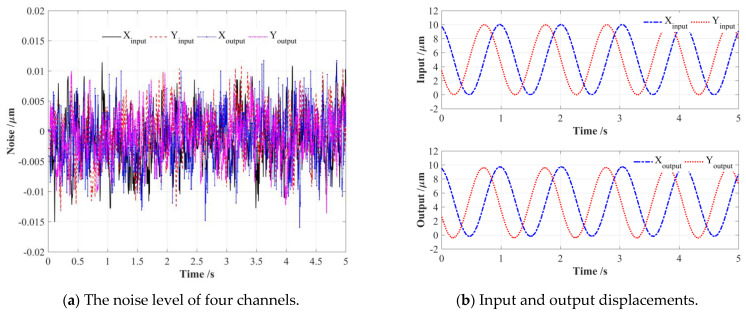
The noise levels of four channels as well as the collected input and output displacements.

**Figure 11 micromachines-14-02043-f011:**
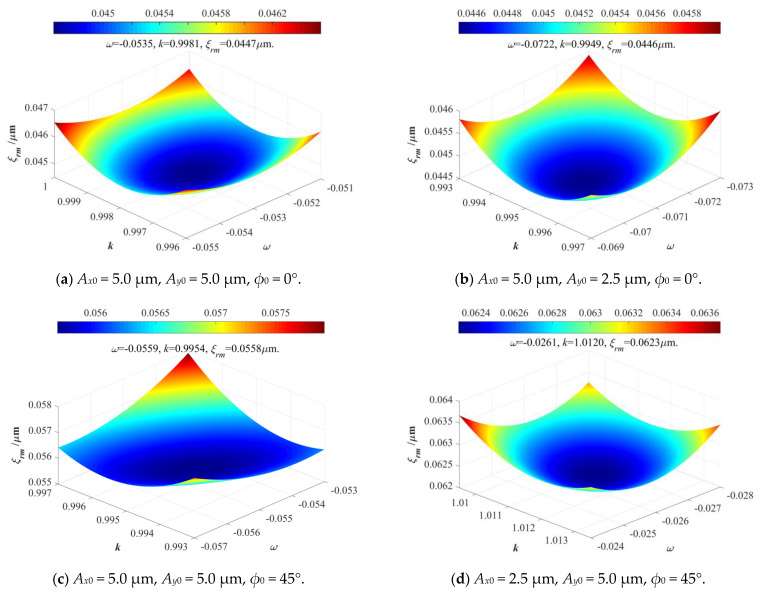
The actual amplification coefficient *κ* and coupling coefficient *ω* under different elliptical parameters.

**Figure 12 micromachines-14-02043-f012:**
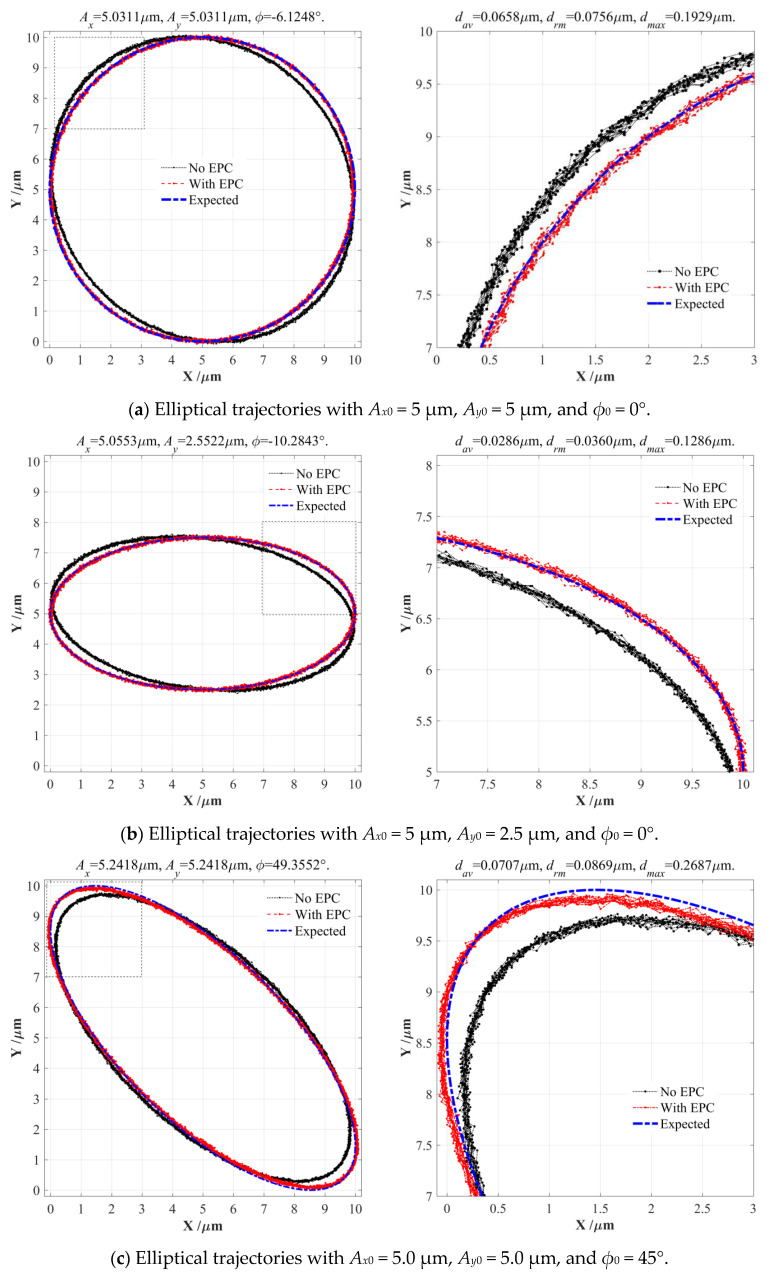
The comparisons of the expected and experimental elliptical trajectories under different motion parameters.

**Table 1 micromachines-14-02043-t001:** The selected dimension parameters of the 2-DOF PDCM.

LSFH	*t*_1_ = 1 mm	*R*_1_ = 2.5 mm	*l*_1_ = 6 mm	*h*_1_ = 4 mm	*d*_1_ = 7.0 mm	*w* = 10 mm
RCFH	*t*_2_ = 1 mm	*R*_2_ = 3.0 mm	*l*_2_ = 10 mm	*h*_2_ = 4 mm	*d*_2_ = 10 mm	*w* = 10 mm

**Table 2 micromachines-14-02043-t002:** The main geometrical parameters of the output elliptical trajectories under different PDC angles.

Geometric Parameters of Output Elliptical Trajectory	Phase Difference Compensation (PDC) Angles Δ*ϕ* (*A_x_* = 10 μm, *A_y_* = 10 μm, *ϕ* = 0°)					
	Δ*ϕ* = 0°	Δ*ϕ* = 2.5°	Δ*ϕ* = 5.0°	Δ*ϕ* = 7.5°	Δ*ϕ* = 8.25°	Δ*ϕ* = 10.0°
Semi-major axis *a*/μm	11.150	10.907	10.659	10.405	10.329	10.331
Semi-minor axis *b*/μm	9.530	9.737	9.940	10.138	10.196	10.147
Relative aspect ratio *λ* = *a*/*b*	1.170	1.120	1.072	1.026	1.013	1.018
Attitude rotation angle *θ*/°	−44.88	−44.82	−44.72	−44.17	−43.42	43.76
Arithmetic average distance *d_av_*/μm	0.5706	0.4335	0.3165	0.2723	0.2627	0.2396
Root mean square distance *d_rm_*/μm	0.6747	0.5299	0.3958	0.2890	0.2676	0.2491

**Table 3 micromachines-14-02043-t003:** The fitted results of the output elliptical trajectories under different PDC angles and motion parameters.

The Fitting Results of Geometrical Parameters of FEA-Based Output Elliptical Trajectories		Motion Parameters and PDC Angles of Input Elliptical Trajectories					
		*A_y_* = 10 μm, *A_z_* = 5 μm, *ϕ* = 0°, Δ*ϕ* = 10.5°	*A_y_* = 5 μm, *A_z_* = 10 μm, *ϕ* = 0°, Δ*ϕ* = 10.5°	*A_y_* = 10 μm, *A_z_* = 10 μm, *ϕ* = 45°, Δ*ϕ* = 8.25°	*A_y_* = 10 μm, *A_z_* = 10 μm, *ϕ* = −45°, Δ*ϕ* = 8.25°	*A_y_* = 10 μm, *A_z_* = 5 μm, *ϕ* = 30°, Δ*ϕ* = 15.0°	*A_y_* = 5 μm, *A_z_* = 10 μm, *ϕ* = −30°, Δ*ϕ* = 12.0°
Semi-major axis length *a*/μm	Expected	10.000	10.000	13.066	13.066	10.372	10.370
	Simulated	10.282	10.274	12.826	14.054	10.501	10.843
	Error	2.82%	2.74%	1.84%	7.57%	1.25%	4.57%
Semi-minor axis length *b*/μm	Expected	5.000	5.000	5.412	5.412	4.175	4.175
	Simulated	5.090	5.094	5.025	6.012	3.614	4.644
	Error	1.81%	1.89%	7.14%	11.09%	13.43%	11.23%
Relative aspect ratio *λ* = *a*/*b*	Expected	2.000	2.000	2.414	2.414	2.484	2.484
	Simulated	2.020	2.017	2.552	2.338	2.906	2.335
	Error	0.99%	0.84%	5.72%	3.17%	16.96%	5.99%
Altitude rotation angle *θ*/°	Expected	0	0	45.000	−45.000	16.845	16.841
	Simulated	−0.400	0.398	44.923	−44.950	17.397	17.018
	Error	---	---	0.17%	0.12%	3.26%	1.05%
Arithmetic average distance *d_av_*/μm		0.1853	0.1840	0.3862	0.8608	0.3907	0.5444
Root mean square distance *d_rm_*/μm		0.1999	0.1973	0.3956	0.8721	0.4597	0.5473

**Table 4 micromachines-14-02043-t004:** The pseudo-decoupling analysis and deviation comparison on different elliptical trajectories.

Motion Parameters of Input Elliptical Trajectories			Error Distances *d_av_* and *d_rm_* Relative to Expected Elliptical Trajectories	
No.	Expected Output Elliptical Parameters	Actual Input Elliptical Parameters	FEA-Simulation	EPC-Calculation
1	*A_x_*_0_ = 10 μm; *A_y_*_0_ = 10 μm; *ϕ*_0_ = 0°.	*A_x_* = 9.7606 μm; *A_y_* = 9.7606 μm; *ϕ* = −8.3732° (Δ*ϕ* = −8.3732°).	*d_av_*^1^ = 2.4589 × 10^−4^ μm *d_rm_*^1^ = 2.4848 × 10^−4^ μm	*d_av_*^2^ = 2.3139 × 10^−5^ μm *d_rm_*^2^ = 2.3224 × 10^−5^ μm
2	*A_x_*_0_ = 10 μm; *A_y_*_0_ = 5 μm; *ϕ*_0_ = 0°.	*A_x_* = 9.7411 μm; *A_y_* = 4.9192 μm; *ϕ* = −10.425° (Δ*ϕ* = −10.425°).	*d_av_*^1^ = 1.7499 × 10^−4^ μm *d_rm_*^1^ = 1.8870 × 10^−4^ μm	*d_av_*^2^ = 2.0978 × 10^−5^ μm *d_rm_*^2^ = 2.2844 × 10^−5^ μm
3	*A_x_*_0_ = 10 μm; *A_y_*_0_ = 10 μm; *ϕ*_0_ = 45°	*A_x_* = 9.2445 μm; *A_y_* = 9.2445 μm; *ϕ* = 38.7511° (Δ*ϕ* = −6.2489°).	*d_av_*^1^ = 1.8040 × 10^−4^ μm *d_rm_*^1^ = 1.9528 × 10^−4^ μm	*d_av_*^2^ = 4.9751 × 10^−5^ μm *d_rm_*^2^ = 5.0771 × 10^−5^ μm
4	*A_x_*_0_ = 5 μm; *A_y_*_0_ = 10 μm; *ϕ*_0_ = 45°	*A_x_* = 4.3924 μm; *A_y_* = 9.4860 μm; *ϕ* = 36.8911° (Δ*ϕ* = −8.1089°).	*d_av_*^1^ = 1.8700 × 10^−4^ μm *d_rm_*^1^ = 2.0069 × 10^−4^ μm	*d_av_*^2^ = 1.6555 × 10^−5^ μm *d_rm_*^2^ = 1.9665 × 10^−5^ μm

**Table 5 micromachines-14-02043-t005:** The experimental analysis and distance deviation comparison under four different elliptical trajectories.

Motion Parameters of Input Elliptical Trajectories			Analyses on Experimental Results	
No.	Expected Output Elliptical Parameters	Actual Input Elliptical Parameters with EPC	Coefficients *ω* and *κ* Distance Deviations *ξ_rm_*	Error Distances with Expected Results
1	*A_x_*_0_ = 5.0 μm; *A_y_*_0_ = 5.0 μm; *ϕ*_0_ = 0°.	*A_x_* = 5.0311 μm; *A_y_* = 5.0311 μm; *ϕ* = −6.1248°. (Δ*ϕ* = −6.1248°)	*ω* = −0.0535; *κ* = 0.9981; *ξ_rm_* = 0.0447 μm.	*d_av_* = 0.0658 μm; *d_rm_* = 0.0756 μm.
2	*A_x_*_0_ = 5.0 μm; *A_y_*_0_ = 2.5 μm; *ϕ*_0_ = 0°.	*A_x_* = 5.0553 μm; *A_y_* = 2.5522 μm; *ϕ* = −10.2843°. (Δ*ϕ* = −10.2842°)	*ω* = −0.0722; *κ* = 0.9949; *ξ_rm_* = 0.0446 μm.	*d_av_* = 0.0286 μm; *d_rm_* = 0.0360 μm.
3	*A_x_*_0_ = 5.0 μm; *A_y_*_0_ = 5.0 μm; *ϕ*_0_ = 45°	*A_x_* = 5.2418 μm; *A_y_* = 5.2418 μm; *ϕ* = 49.3552°. (Δ*ϕ* = 4.3552°)	*ω* = −0.0559; *κ* = 0.9954; *ξ_rm_* = 0.0558 μm.	*d_av_* = 0.0707 μm; *d_rm_* = 0.0869 μm.
4	*A_x_*_0_ = 2.5 μm; *A_y_*_0_ = 5.0 μm; *ϕ*_0_ = 45°	*A_x_* = 2.3796 μm; *A_y_* = 4.8973 μm; *ϕ* = 42.1744°. (Δ*ϕ* = −2.8256°)	*ω* = −0.0270; *κ* = 1.0120; *ξ_rm_* = 0.0623 μm.	*d_av_* = 0.0523 μm; *d_rm_* = 0.0634 μm.

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
