# Peer review of "A Novel Type of Pseudo-Decoupling Method for Two Degree-of-Freedom Piezoelectrically Driven Compliant Mechanisms Based on Elliptical Parameter Compensation"

_micromachines, 2023, doi:10.3390/mi14112043_

Round 1

Reviewer 1 Report

Comments and Suggestions for Authors

The authors proposed a new pseudo-decoupling method based on phase difference compensation to improve the trajectory tracking precision of widely-used EVM apparatus. And an amplification coefficient and a coupling coefficient are defined to mathematically model the EPC-based method. By comparison with the FEA-simulated, EPC-calculated and expected output elliptical trajectories, the proposed pseudo-decoupling methods can effectively improve the trajectory tracking precision of conventional 2-DOF PDCM. Finally, the effectiveness of the proposed model is verified by comparison with the experiments. In general, this woke is solid and convincing. But some issues need to be addressed.

1. In Fig.1, the local representation “w” is not analyzed and discussed in the main text. I suggest the authors to delete it.

2. In Page 5, the authors mentioned that “the input and output elliptical trajectories generated by 2-DOF PDCMs will be numerically simulated and analytically compared with finite element analysis (FEA) method, respectively, as illustrated in Fig 3”. But the legends of Fig 3 “Expected and Simulated” are not consistent with this description. The authors should rewrite these sentences or correct the legend of Fig 3. Besides, the title of the subgraph in Fig 3 are basically the same. I suggest the authors to simplify it.

3. The Tables 2 and 3 span two pages. The authors should rearrange them.

4. The Equation (2) is too long. The terms “xin(t)” and “yin(t)” are described in Equation (1). The authors should shorten it. And the description of “x(t)out”, “y(t)out”, “x(t)in”, “y(t)in” should be consistent with those in the Equations.

Comments on the Quality of English Language

The grammar of the text needs to be checked. For example, in Page 2, the layout of the references 6 and 7 is not correct. They should be consistent with other references. And the description of “… have be potentially employed in …” should be “… have been potentially employed in …”. The authors need to double-check this manuscript to avoid syntax errors.

Reviewer 2 Report

Comments and Suggestions for Authors

The static input and output models of the two-degree-of-freedom piezoelectric driven flexible mechanism are established. The phase hysteresis, displacement amplification and coupling effects of the two-degree-of-freedom piezoelectric driven flexible mechanism are considered. The quantitative relationship between the input ellipse parameters and the output ellipse parameters is established by introducing compensation Angle, amplification coefficient and coupling coefficient. The three parameters are optimized based on the principle of minimum error distance between expectation and finite element simulation elliptic trajectory. The experimental results prove the effectiveness of the model and provide an effective control method for more accurate elliptic trajectory control.

In order to improve the paper, it is suggested that the authors make necessary explanations and revisions to the following questions.

1. Based on the application background of elliptical vibration machining, the static model is obviously unable to accurately predict the dynamic process. What are the effects of the dynamic inherent characteristics of the flexible mechanism? What are the amplitude-frequency response characteristics of the flexible mechanism when the elliptic trajectory vibrates at different frequencies?

2. How is the value range of amplification coefficient and coupling coefficient selected and reduced to the range described in the paper? Is it possible that the solution is locally optimal?

3. The second and third parts introduce two pseudo-decoupling methods respectively, which are logically side by side, and the research process is progressive. It is suggested to modify the title of the second part as "Pseudo-decoupling Method based on phase compensation".

4. The abbreviations of flexure hinge, especially those in Figure 1 are not uniform and are not explained in the text,  which should be unified with the text.

5. It is recommended to briefly explain how to define and calculate the two error distances of the elliptic trajectory.

Reviewer 3 Report

Comments and Suggestions for Authors

The authors of this manuscript present a method to improve the precision of trajectory tracking of 2-DOF piezoelectrically driven compliant mechanisms with a pseudo-decoupling method based on phase difference and elliptical parameter compensation. The approach is comprehensive and interesting. However, this paper should be revised before it can be published.

Main questions or remarks are followings:

1)    Figure 12c: There are still slight deviation between expected and “with EPC” results. What would be the reason? Please indicate it clearly.  

2)    How about the hysteresis of the piezoelectric actuators? Did not they show any hysteresis? Have you taken hysteresis in to account at your approach?

3)    Can your approach work at higher frequencies as well? What is the typical frequency of the elliptical or sinusoidal paths? Are they excited with 1 Hz? 

Comments on the Quality of English Language

Moderate editing of English language required

Round 2

Reviewer 3 Report

Comments and Suggestions for Authors

I have no further questions related to the revised version of the paper.

Comments on the Quality of English Language

Minor changes required.